# Driving Behaviour Estimation System Considering the Effect of Road Geometry by Means of Deep NN and Hotelling Transform

Felipe Barreno [1], Matilde Santos [2,*] and Manuel Romana [3]

1 Faculty of Informatics, Complutense University, 28040 Madrid, Spain; febarren@ucm.es
2 Institute of Knowledge Technology, Complutense University, 28040 Madrid, Spain
3 E.T.S.I. Civil Engineering, Polytechnique University of Madrid, 28040 Madrid, Spain; manuel.romana@upm.es
* Correspondence: msantos@ucm.es

**Abstract:** In this work, an intelligent hybrid model is proposed to identify hazardous or inattentive driving manoeuvres on roads, with the final goal being to increase and ensure travellers' safety and comfort. The estimation is based on the effects that road geometry may have on vehicle accelerations, displacements and dynamics. The outputs of the intelligent systems proposed are how the type of driving can be characterized as normal, careless or distracted. The intelligent system consists of an LSTM (Long Short-Term Memory) neural network in a first step that distinguishes between normal and abnormal driving behaviour and then a second module that classifies abnormal forms of driving as aggressive or inattentive, with the latter implemented with another LSTM, a CNN (convolutional neural network) or the Hotelling transform. They are applied to some of the characteristics of vehicle dynamics to estimate the driving behaviour. Smartphone inertial sensors such as GPS, accelerometers and gyroscopes are used to measure these vehicle characteristics and to identify driving events in manoeuvres. Specifically, the critical acceleration due to the influence of the road geometry can be measured with inertial sensors, and then, this road acceleration with the lateral acceleration allows us to estimate the driver's perceived acceleration. This perceived acceleration affects the driving style and, consequently, the estimation of the appropriate speed to travel on that road. There is use of both a traditional two-lane and a motorway route located in the Madrid region of Spain. Driving behaviour is determined by considering how changes in road geometry may affect one's driving style and, consequently, the estimation of the proper speed. The results obtained with some of the proposed configurations of the intelligent hybrid system reach an accuracy of 97.21% in detecting dangerous driving or driving with a certain risk. This could allow generating real-time alerts for potentially dangerous or inattentive manoeuvres, leading to safer and more appropriate driving.

**Keywords:** convolutional neural networks; LSTM neural networks; Hotelling transform; ADAS system; smartphone sensors; driving behaviour; roads; vehicles; Industry 4.0

## 1. Introduction

The analysis of the behaviour of vehicle drivers is of increasing interest, since it can be used for a wide variety of purposes. In the automotive field, the detection of inattentive or aggressive driving behaviours is essential in order to improve safety or to implement control changes in partially autonomous vehicles. Today, the increased use of vehicles is having several negative effects, among them, heavy traffic, crashes, injuries, fatalities and economic losses. Human errors due to factors such as fatigue, alcohol, recklessness or carelessness are the main causes of most accidents. However, this situation could be alleviated since the information on the state and attitude of the driver, the environment and the vehicle can improve safety on the roads, especially when an abnormal or unforeseen situation occurs [1].

To protect the safety of the driver and other road users, it is crucial to recognize various driving habits that may pose dangers, such as distraction or drowsiness, according to the

automotive industry and traffic enforcement [2]. Categorizing and observing the driving behaviour are essential for risk evaluations and also for pricing in auto insurance companies [3].

Road infrastructure geometry and state may both influence driving safety through their perception by the drivers. Driver behaviour as well as road geometry and condition are the most important factors in the safety and comfort of the trip [4,5]. Specifically, the critical acceleration due to the influence of the road geometry can be measured with inertial sensors and then used with the lateral acceleration to estimate the driver's perceived acceleration. This perceived acceleration affects the driving style and, consequently, the estimation of the appropriate speed to travel on a specific road.

In addition, drivers' conditions are not constant and may change since each user perceives driving situations differently, even with no time or climate changes, according to the vehicle he or she is driving, road curvature, section curviness, etc. The subjective views of the driver must be taken into account when analysing their driving [6]. Vehicle features, such as accelerations and trajectories, can be used to characterize driving behaviour. For example, in [7], inertial vehicle units (IMU) allow for the construction of the driver profile including driver braking information and cornering manoeuvres.

This paper aims to investigate whether a certain driving behaviour can be considered unusual. Using machine learning techniques, normal or abnormal manoeuvres may be identified. The proposed model predicts the behaviour of a driver on the road based on the speed at which it is moving and the exerted force on the steering wheel. One of the features studied is how drivers react when their chosen speed, which they believe to be adequate, is not the appropriate one for the specific geometry of the road. The intelligent estimation model designed for the characterization of driving manoeuvres includes the road geometry considering characteristics such as the onward and lateral perceived vehicle accelerations [8]. This approach's primary contribution is that it tackles driving anomalies that arise from the driver's perception of the vehicle accelerations that are caused by road geometrical characteristics. In certain cases, the authorized driving speed may be excessively high, making the given road geometry dangerous.

The intelligent system presented in this research consists of two modules: one uses a deep neural network based on the Long Short-Term Memory (LSTM) recurrent network to identify which driving movements are anomalous; the second module has been implemented with different technologies and uses the output of the previous model as input to identify whether the abnormal manoeuvre corresponds to aggressive behaviour or indicates distraction on the part of the driver. In this second phase, a Hotelling transform (HT)-based classifier, a convolutional neural network (CNN) and another LSTM network have been tested.

The hybrid system that best distinguishes between the three driver profiles is the LSTM-HT configuration, with an accuracy in identifying each driving profile of up to 97.21% and an F1-score of 98.38%.

The main contribution of this work consists of incorporating the geometry of the road as another variable to evaluate its influence on the driving style or on a manoeuvre, something that is not usually taken into account. Furthermore, it is carried out indirectly but through real inertial measurements taken by low-cost devices, with which the acceleration perceived by the vehicle's passengers is obtained without the need to have direct information on the actual state of the infrastructure.

This study has been carried out on a two-lane road and a dual carriageway freeway in Madrid (Spain). The results are promising, since the information obtained can help establish a more safety-conscious driving style, achieving smooth driving. This approach also supports the use of GPS data from smartphone devices and inertial sensors. This affordable device is intended to assist drivers in preventing collisions by identifying the type of conduct and differentiating between careless, normal and inattentive behaviour.

The structure of this document is as follows. Some similar studies are commented on in Section 2. The vehicle dynamics that make up the proposed classification model's

framework are defined in Section 3. The model based on using real data that estimates when a motion is abnormal is described in Section 4. Section 5 presents the main findings. Discussion of results is presented in Section 6 and conclusions are described in Section 7. The article is concluded with conclusions and potential areas for future research.

## 2. Background and Related Works

Driving behaviour can be understood as the practices, manner and attitudes of a typical user whilst driving, which could be categorized into various styles: normal, safer driving, aggressive driving, inattentive driving, drowsy driving or driving under the influence of alcohol, among others [2]. In [9], safe behaviour at the wheel is defined as the usual everyday behaviour of an individual user, while non-normal driving behaviour is the infrequent behaviour of an individual driver whenever it is affected by mental or physical factors. Normal driving is related to a behaviour that avoids risky reactions and abrupt manoeuvres. Driving without reckless manoeuvres is categorized as a careful or safe driving style [10]. When the driver attempts to minimize travel time it can be considered as reckless driving, since it very often carries inappropriate anomalous and sudden vehicle speed changes and lateral positions of the vehicle, dangerous lane changing and accelerating and decelerating rapidly [11,12]. A driver's intermittent inattention to the driving task and its necessary attentive actions can result in a pattern of distracted driving, which is often followed by the driver's rapid and sudden reaction to correct the vehicle's position. This results in a driving style of an instantaneous and irregular nature. Inattentive or distracted driving is characterized as the deviation of focus away from critical activities necessary for safer driving. It is also related to a driver's behaviours when exhausted or fatigued [13].

In the literature, the prediction of driving patterns using computational methods has been addressed with different techniques, and this fuels the interest of the topic and the suitability of some strategies. Indeed, one of the preferred means of the analysis of road behaviour is to examine manoeuvres, as these occurrences can provide helpful data on the driving forces involved. Neural networks have been applied to this task.

Because deep learning algorithms can automatically understand the temporal dependencies in a time series, they are emerging as a viable and affordable alternative for modelling driving [14] and, thus, for identifying normal and abnormal patterns. Long short-term memory (LSTM) networks constitute a good architecture for computing data time series [15]. For instance, [16] propose a method for anomalous behaviour recognition at the wheel based on a full convolutional LSTM network. According to [17], the replicating NN (RNN) and LSTM are used to classify abnormal driving styles. Using convolutional neural networks, the authors in [18] describe typical, aggressive, distracted, sleepy and intoxicated driving behaviours. In [19], anomalous driving behaviour is identified by the use of NN-stacked short-term memory.

Driver behaviour detection and characterization have been facilitated by the widespread use of multivariate statical process models that consider Principal Component Analysis (PCA) as well as Hidden Markov Models (HMM) in pattern categorization [20]. A machine learning technique is used to model and identify driver behaviour in [21], with the information set built by combining steering wheel angle sensor, brake pedal, speedometer, accelerator, gear engaged and GPS. This set is used as input to the HMM. In [22], an adaptive assistance system has also been developed to determine or predict driver behaviour using HMM.

In [23], the authors propose combined process monitoring metrics for early detection of failures by making use of PCA. In [24], it is proposed to identify the drowsy state of a driver by observing the heart rate variability without labelling it manually. A classification model for driving behaviour is implemented using neuro-fuzzy methods and weighted on PCA in [25]. In [26], the relationship between driving behaviour and reckless driving is obtained using naturalistic driving data with K-means cluster analysis, PCA methods and regression

models. A deep learning model is presented to study the complex interactions between the road environment and driver behaviour through a graphical representation in [27].

In order to improve the safety management of road infrastructures, a method is proposed in [28] to evaluate the coherence of current road layouts by examining the geometric characteristics, theoretical speeds and operating speeds of drivers under various environmental and flow conditions. The study focuses on the ANAS SpA-managed road network in the Veneto Region, for which the theoretical design speed profile, the curvature graph and the reconstruction of the road axes geometry have been obtained. In a similar way, the authors of [29] offer a technique as a useful tool for comprehending safety issues and formulating enhancements to infrastructure inspection protocols to modify them for use on secondary and country roads. The investigation was conducted on a section of the SS7 Appia state road in Lazio, Italy.

In this paper, unlike the mentioned works found in the literature, the novel contribution is achieved by addressing the effect of road geometry on a driver's driving behaviour. In this context, normal driving is referred to as safe driving. Careless manoeuvres are considered as aggressive or reckless driving. Finally, inattentive manoeuvres include drowsy manoeuvres and distracted driving because this behaviour can be due to smartphone distractions or radio interaction during the trip.

## 3. Vehicle Dynamics-Based Approach and Road Geometry Effects

The forces defining the dynamics of the vehicle affect it [30]. Moreover, the variables that affect vehicle dynamics—like tire pressure or vehicle weight, for instance—may change over time. Furthermore, the road's geometry is planned so that, at a certain speed, the driver and passengers will feel comfortable and secure enough.

The road geometric characteristics define the design speed $V$ (m/s) of a section in a road, irrespective of the maximum speed permitted by the traffic rule; it calculated as follows (FOM/273/2016):

$$V^2 = 127R(f_t + \frac{\rho}{100}) \tag{1}$$

where $f_t$ is the highest coefficient of cross friction, $\rho$ (%) is the road's transverse slope, and $R$ stands for curvature radii (m). Linear acceleration is the acceleration measured in a forward direction that runs parallel to linear speed [31]:

$$\frac{V^2}{gR} = \rho + f_t \tag{2}$$

$$a_{road} = g(\rho + f_t) \tag{3}$$

where $g$ (m/s$^2$) is the acceleration caused by gravity, and $a_{road}$ (m/s$^2$) is the critical acceleration caused by the influence of the road. This speed limit applies when accelerating through a horizontal curve to maintain both comfort and safety while driving.

Since determining the maximum friction coefficient and the cross slope may be challenging, the variable $a_{road}$ can be measured using the GPS and IMU. The acceleration as a function of road characteristics can also be given as follows [32]:

$$a_{road} = |\omega| \cdot v_l \tag{4}$$

In (4), $v_l$ (m/s) represents the linear vehicle speed, and $\omega$ (rad/s) is the yaw or swerve rate. As per [33], the driver who exhibits risky driving conduct is considered to have experienced the following lateral acceleration:

$$a_p = |a_m| - a_{road} \tag{5}$$

where $a_m$ (m/s$^2$) is the lateral acceleration, and $a_p$ (m/s$^2$) is the driver's perceived acceleration. The latter is in line with the "feeling" that the driver experiences as a result of the

geometry of the route. Equation (6) defines the rate of change in longitudinal acceleration, which is also known as the gradient of acceleration or jerk in a given $\Delta t$.

$$Jerk = \frac{d\mathrm{a}(t)}{dt} = \frac{\mathrm{a}(n) - \mathrm{a}(n-1)}{T}; t = nT \tag{6}$$

Acceleration peaks determine the 'safe' margin and relate to sudden or erratic driving manoeuvres for a safety driving style.

To summarize, the influence of the road geometry is estimated from the real lateral acceleration on the vehicle measured by inertial sensors using the expressions (3) and (5) [30]. The geometric characteristics of the road, radius of curvature and cross slope, are included indirectly in the $a_{road}$ acceleration [31], which can be estimated using the linear vehicle speed and the yaw rate [32]. Then, the acceleration that the vehicle's passengers feel during a trip is obtained [33]. For example, if a small radius curve is taken at considerable speed, passengers get the sensation of moving out of the curve, which is measured by perceived acceleration.

Thus, the variables used in the behaviour estimation model are longitudinal and lateral acceleration, vehicle speed and vehicle angular velocity. Jerk, the angle of the car in relation to the centre line, and the distance between the vehicle position and the centre line will also be used. These are obtained from the vehicle driving dataset used. The driver's perception of acceleration and the jerk are extracted from these to characterize and label the training dataset. The vector of features that will be used in the intelligent models to identify different driving styles will be based on these variables.

## 4. Materials and Methods

### 4.1. Intelligent Model of Driving Behaviour Estimation

Every driver perceives safe speed in a completely different way. Moreover, each driver can act similarly, and even have an identical driving trip, for instance, based on the degree of urgency that he or she feels. Also, each driver has a different appreciation of the driving style applied to certain driving conditions, such as heavy traffic or bad weather conditions.

The estimation model developed here addresses some of these subjective driver perceptions which are extracted from basic measurements of inertial sensors placed in a vehicle. The intelligent system is made up of two interconnected modules (Figure 1). The intelligent estimation model is divided into two stages to identify the manoeuvre type: normal, aggressive or inattentive. The first module is implemented with a deep neural network using a recurrent long short-term memory (LSTM) neural network. This module receives four input features from the vehicle, namely, linear speed, longitudinal and lateral accelerations and angular velocity. The second module is fed with the acceleration rate processed from longitudinal acceleration, that is, jerk obtained by longitudinal vehicle movement, position of the vehicle compared to the lane centre and the angle of the vehicle compared to the curvature lane. Previously, a sensor fusion layer was implemented to adapt inertial measurements from vehicle and post-processed signals. The outputs of the system are either normal driving, obtained directly from the first LSTM module, or abnormal, i.e., reckless or inattentive driving, as a result of the second proposed classifier.

With the inertial measurement unit (IMU) of the vehicle, it is possible to obtain the longitudinal and lateral acceleration, the angular speed and the velocity of the vehicle. From them, it is possible to estimate the perception that the driver has while driving, that is, the orthogonal force that moves the vehicle towards the outside of the road when passing through a curve at a certain speed (Equations (2)–(6)). Thus, we obtained the effects of the characteristics of the geometry of road that are implicit during the movement of the vehicle.

The public UAH-DriveSet [34] has been used in this work. These naturalistic driving data were collected by the "DriveSafe" application [32] in different environments and conditions, and they include information on different test drives.

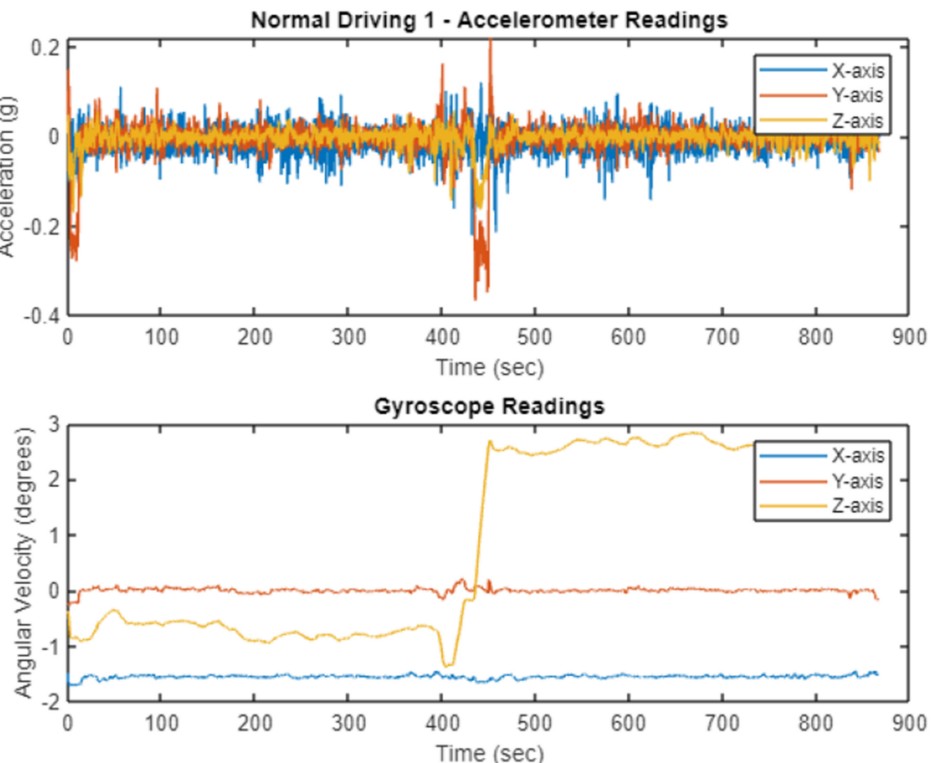

**Figure 1.** Accelerometer and gyroscope readings on the A-2 highway road (Spain) [34].

More than 500 min of recorded driving time from six drivers operating various cars—gasoline, diesel and electric—are included in this dataset (Table 1). Three distinct behaviours were taken into consideration: aggressive, sleepy and normal. They all travel prearranged routes on two distinct types of roads: a secondary road (M-100, Madrid region, Spain) that runs for 16 km and a freeway (A-2, Spain) that runs for 25 km. Participants were asked to make three journeys, each with a different type of behaviour. In the normal style, participants acted calmly and politely. In the aggressive run, they acted aggressively and with anger. Operating a motor vehicle while mentally impaired owing to sleep deprivation is known as drowsy or sleepless driving.

**Table 1.** Users and vehicles used in UAH-DriveSet.

| Driver | Genre | Age Range | Vehicle | Fuel Type |
|--------|--------|-----------|----------------------|-----------|
| D1 | Male | 40–50 | Audi Q5 (2014) | Diesel |
| D2 | Male | 20–30 | Mercedes B180 (2013) | Diesel |
| D3 | Male | 20–30 | Citroën C4 (2015) | Diesel |
| D4 | Female | 30–40 | Kia Picanto (2004) | Gasoline |
| D5 | Male | 30–40 | Opel Astra (2007) | Gasoline |
| D6 | Male | 40–50 | Citröen C-Zero (2011) | Electric |

The database contains real-time raw measures from inertial sensors (accelerations and gyroscopes) and smartphone GPS [34]. The files include details about the vehicle's speed, location on the road, distance from the automobile in front, the kind of road it is on, and its angle relative to the road. The data are acquired continuously on a conventional road located in Madrid (Spain). The vehicles run two routes, one that is mostly a "motorway" type road, consisting of between 2 and 4 lanes in each direction and about 120 km/h maximum allowable speed. The other route covers mostly a "secondary" type road, consisting mainly of 1 lane in each direction and about 90 km/h maximum speed. Each driver (D1–D6) made three trips on the motorway road (outbound and return, about 25 km each), simulating each of the three behaviours (normal, inattentive, aggressive), and four

trips on the secondary road (outbound, about 16 km each), consisting of exit as normal, return as normal, exit as aggressive and return as drowsy. This database collected by the DriveSafe application [32] has a significant amount of information from each route, including raw measurements and processed signals (semantic information), such as the image captured by the rear-view camera. The dataset is divided into folders for each driver, and the files contain several variables arranged in columns. The first column always represents a 'timestamp' indicating the number of seconds elapsed. The elapsed time in seconds since the start of the route enables synchronization between the different files and the corresponding video.

Figure 1 shows, as an example, some dynamic characteristics of the car (Audi Q5 D1 driver) on a stretch of the A-2 highway road [34]. This driving behaviour describes normal driving. The top figure shows the measurement of the acceleration (in *g*, gravity); the bottom figure shows the gyroscope measurements for each x, y and z coordinate.

*4.2. Feature Selection*

The data are collected at multiple sampling rates. Thus, a sensor fusion layer has been implemented to solve this. This sensor fusion layer combines sensor information from a variety of sources in such a way that the derived uncertainty in the outcome data is less significant than it would be if these samples were obtained separately. The GPS and inertial sensor have sampling times of 1 Hz and 10 Hz, respectively. To ensure synchronization, the GPS data were considered constant until the next sample time, and the corresponding accelerometers data were sub-sampled at 1 Hz.

A typical behaviour at the wheel, here and in earlier research, is characterized using longitudinal acceleration, lateral acceleration, angular velocity (which determines if the vehicle is developing a tendency to turn around the vertical axis) and vehicle speed were identified as the most important characteristics.

The linear speed $v_l$ at which a vehicle travels is measured in km/h (obtained by the GPS). The vehicle is in the centre of the road. The relative angle ($\varphi$) of the vehicle to the centreline (degrees), that is, the angle between the car and the road centreline, indicates how close the car will be to the edge of the road. The position distance of the vehicle from the centre line is $X_d$. As a result, these variables indicate the vehicle's lane cantering, and the jerk (*z*-axis) indicates sudden changes in longitudinal acceleration. This will lead to the awareness of any unusual steering wheel movements.

The input variables for the first system (the LSTM-based subsystem, to be used to identify any unnormal behaviour) and to the second system (the deep neural network-based and Hotelling transform-based models to estimate the aggressive or inattentive behaviour) are detailed in Table 2.

**Table 2.** Description of features used to intelligent model.

| Features | Descriptions | Units |
|---|---|---|
| $a_z$ | Longitudinal acceleration | g |
| $a_y$ | Lateral acceleration | g |
| $w_z$ | Angular speed Yaw rate | rad/s |
| $v_l$ | Linear velocity | km/h |
| $Jerk_z$ | Longitudinal jerk | $m/s^3$ |
| $X_d$ | Position of the vehicle in relation to centre | m |
| $\varphi$ | Angle of vehicle to lane curvature | ° (degrees) |

It is Although it is difficult to define, it can be accepted that typical or normal driving describes a driver that steers clear of potentially dangerous situations and reactions, in contrast to aggressive, distracted, careless, sleepy or inebriated driving that usually entails abrupt and unusual changes in speed, improper maintenance of the vehicle's lateral position, a quick response to rectify vehicle position, risky lane changes, and rapid acceleration and deceleration [18].

While the sideways standard deviation is associated with driver's steadiness, the longitudinal speed and acceleration standard deviation can reveal the consistency of the driver's longitudinal control [35]. Metrics that are mainly used to specifically assess comfort and safety during driving manoeuvres are the mean and standard deviation of variables such as acceleration or jerk because humans are more sensitive to rapid changes in acceleration, as stated in [36]. Equations (7) and (8) express that a valuation outside of the expected value of the variables that measure the stability of the longitudinal and lateral control of the vehicle is considered a driving behaviour different from what is established as normal [36].

Consequently, considering forward acceleration and also driver-perceived lateral acceleration as explained below (7), anomalous or abnormal driving behaviour was labelled.

$$\overline{X}_a = \begin{cases} a_z < \mu_{a_z} & x = normal \\ a_p < \mu_{a_p} & x = normal \\ otherwise & x = abnormal \end{cases} \tag{7}$$

where $\overline{X}_a$ is a characteristic vector made up of features from Table 3, $\mu_{a_z}$ is the average of the longitudinal acceleration, $\mu_{a_p}$ is the average perceived lateral acceleration, and $x$ is the class that indicates the type of driving. Aggressive and inattentive driving behaviours samples were labelled considering events on longitudinal jerk acceleration performed by the driver as follows (8):

$$\overline{X}_b = \begin{cases} Jerk_z > \mu_{Jerk_z} & x = reckless \\ otherwise & x = inattentive \end{cases} \tag{8}$$

**Table 3.** Input features of Hotelling transform-based classifier.

| Features | Descriptions | Units |
|---|---|---|
| $Jerk_z$ | Longitudinal jerk | $m/s^3$ |
| $X_d$ | Vehicle position to lane centre | m |
| $\varphi$ | Vehicle angle to curvature lane | ° (degrees) |

Again, $x$ denotes the class, $\mu_{Jerk_z}$ is the average longitudinal jerk, and $\overline{X}_b$ represents the characteristic vector consisting of features from Table 3. The threshold to identify erratic or aggressive acceleration for labelling data samples is set considering mean accelerations and jerk values. The thresholds for sudden acceleration and sudden braking could be defined as a function of the driving style depending on the individual driver and are therefore characterized as the mean value of jerk values [16].

Driver-style characterization is based on the following. Considering an acceleration rate that is conservative and comfortable, it is within the limits of the driver's ability to stay on the road and keep steering control during the under-braking manoeuvres. If the vehicle speed is too fast when cornering, the lateral acceleration will increase, and the steering will be poor. The user's perception of lateral acceleration is crucial because it conveys the driver's feelings while operating a vehicle, so the vehicle dynamics—including speed, angular velocity and longitudinal and lateral acceleration—allow for an indirect analysis of the impact of the road geometry. This can indicate that the driver is exceeding the posted speed limit.

In other words, from the vector of the vehicle dynamics characteristics and the road geometry features from the displacement of the vehicle (Table 2), the acceleration perceived by the driver can be obtained during turning manoeuvres (Equation (5)) and therefore implicitly includes the impact of the road geometry.

### 4.3. Deep Long Short-Term Memory Neural Network for Identification

Long short-term memory recurrent neural networks are used extensively in applications of pattern detection as a supervised algorithm [37]. They are good at processing

time series data due to memory capacity, and the output of the current cell depends on the current input and the output of the previous cell [38]. In this work, a recurrent neural network LSTM is applied to the detection of anomalous driving manoeuvres [39] using the actual measurements collected by in-vehicle smartphone sensors during driving.

There is a total of 275,220 samples that contain all of the drivers' profiles, of which, 41.630 belong to driver D1. The data were initially randomly divided into 50% for training and the other 50% for testing, making use of all driving profiles. To improve the models, other experiments were carried out in which the driving profile D1 was used to train the model and the rest of the profiles for testing.

In the database, 90,464 out of the 275,220 data points have the classification "abnormal driving". To deal with severe value variations in the network weights, the z-score is utilized to homogenize the characteristics when the various aspects are gauged.

This first module is a deep learning neural network with one input layer with 4 neurons, 1 for each feature, and 3 hidden layers with 125, 100 and 100 neurons, respectively. These layers have a dropout layer to prevent overfitting. A fully connected layer is built where a linear transformation is performed to the input vector through a weight matrix, allowing each input vector to influence each output. In the softmax layer, the output values of the fully connected layer are normalized between 0 and 1. Lastly, the classification layer yields the result.

As said, two training approaches are presented, one which uses driver D1 data (Table 1) for training and another selecting randomized training and test data of all the driving profiles.

Figure 2 shows the deep neuronal network structure based on LSTM cells of the first module. Some experiments were carried out, and the best results were achieved with the following parameter configuration of the LSTM neural network.

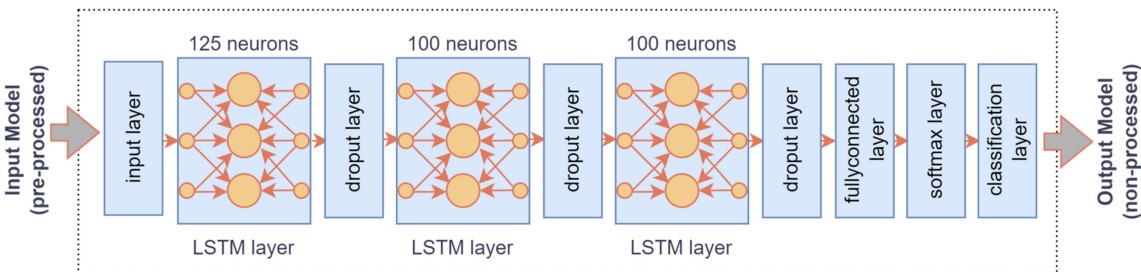

**Figure 2.** Proposed deep LSTM neural network for the initial classification.

The output of this LSTM network identifies normal driving vs. any other driving that reflects anomalous manoeuvres. These results will be used as input in the next module that will discriminate between types of abnormal driving style.

### 4.4. Classifier Based on Hotelling Transform (HT)

The second module of the driving style identification system is implemented with several techniques, among others, with the Hotelling transform (HT). This is an implementation of the well-known PCA method, which is in charge of the decomposition of a multivariate dataset into a set of successive orthogonal components that can explain the variance. The PCA approach converts the original space input information to the processed space only with its uncorrelated information [40]. This processing is carried out by the Hotelling matrix $U$. The HT matrix $U_k$ is calculated for every class $k$, so in our case, $k = 2$ {reckless, inattentive}.

Generally, PCA is sub-divided into two phases [41]. The first, also called the training phase, is performed offline and works with a set of previous measurement vectors containing the information corresponding to a given class or pattern, in this case, from the driving profile of first driver (D1) (Table 1), which is labelled based on (9) as the pattern driving profile. A set of data $\bar{x}_k$ is created, and the zero-mean data set $\varphi_k$ is applied to obtain the

Hotelling matrix $U_k$ that turns the original space to the transformed space. The eigenvectors associated with the eigenvalues of the matrix of covariance $S_k$ generate the matrix $U_k$ of the pattern dataset. This training phase uses a number of N measurements from the input dataset which represent different driving manoeuvres for class $k$. Each characteristic vector is defined in terms of $\bar{x}_{mk}$ (9) being $\bar{x}_{mk} \in \mathrm{R}^3$ (longitudinal jerk, distance to lane centre and curvature angle to lane centre). Hence, the vector of means $\varphi_k$ (10), the zero-mean vector $x_{mk}$ (11) and the matrix of covariance $S_k$ (12) are calculated.

$$\bar{x}_{mk} = [Jerk_z, X_d, \varphi]_{mk} \quad m = \{1, \ldots, N\}; \quad k = \{reckless, inattentive\} \tag{9}$$

$$\varphi_k = \frac{1}{m} \sum_{m=1}^{m} \bar{x}_{mk} \tag{10}$$

$$x_{mk} = \bar{x}_{mk} - \varphi_k \tag{11}$$

$$S_k = \frac{1}{m} \sum_{m=1}^{m} (x_{mk})(x_{mk})^T \tag{12}$$

The second phase, the online classification, uses the measurement vector formed by the data obtained at each time instant from the reading of the inertial measurement unit. Using the transformation matrix $U_k$, the received mean value of the system used to obtain a zero mean vector $x$ is projected into the transformed space (one per class $k$) to obtain the new vector of transformed characteristics $y_k$ (13).

$$y_k = U_k^T x \tag{13}$$

$$\bar{x}_k = U_k y_k \tag{14}$$

Subsequently, the reconstructed vector $\bar{x}_k$ for each class k is obtained using (14). The reconstructed data $\bar{x}_k$ derived from the original $x$ differ according to degree of similarity among the new data $x$ and the data used to produce the transformation matrix $U_k$. This is called the error of reconstruction $e_k$ (15) and is calculated for each class $k$ by the Mahalanobis distance among $x$ and $\bar{x}_k$.

$$e_k = \left(x - \bar{x}_k\right)^T S_k^{-1} (x - \bar{x}_k) \tag{15}$$

Then, the minimal error of reconstructing $e_k$ qualifies the input vector $\bar{x}$ as a member of class $k$.

To summarize the procedure, the Hotelling transformation $U$ is determined at N training vectors. Then, given a measurement vector, this is transformed to the feature space through the given transformation matrix $U$. The vector is then recovered from the transformed space via the inverse transformation. Finally, the distance between the original vector of each class to the recovered vector is called the recovery error. Thus, the smallest distance to each class determines the classification or membership of a given class. Figure 3 shows the process described for classifying a set of manoeuvres.

The methodology of the application of the HT, as represented in Figure 3, is as follows: from the model input information, a set of standard classes is generated, and for each data forming the class, the zero mean is extracted, and then, the covariance of the class is calculated. From the eigenvectors of the covariance, the transformation matrix is formed. When data to be classified arrives, the zero mean of each class is extracted from these data; the linear transformation is calculated to project the data to an uncorrelated space and then reconstructed to the original space of each class. Once the data are in the original space, the construction error is calculated using the Mahalonobis distance from the recovered data to the class. The data are classified to a certain class if the error is minimal.

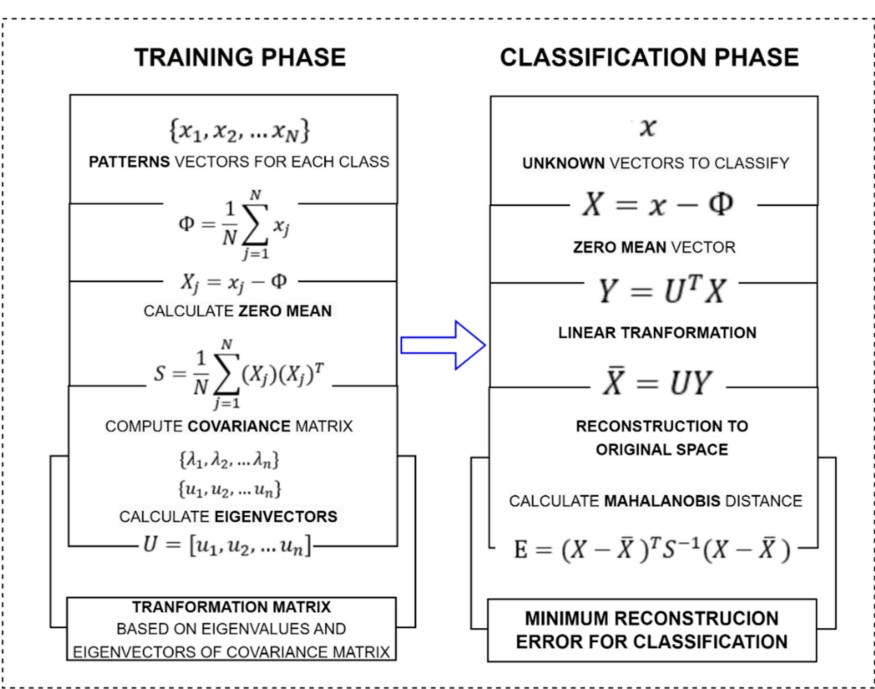

**Figure 3.** Hotelling transform-based proposed classifier.

The input features of the Hotelling transform-based classifier to identify reckless or inattentive manoeuvre from aggressive driving behaviour are shown in Table 3, and the whole system is shown in Figure 4.

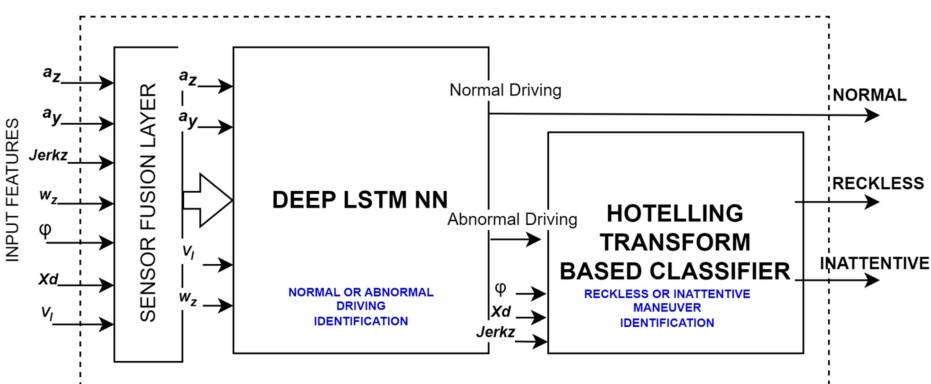

**Figure 4.** Proposed driving behaviour model using Hotelling transform.

### 4.5. Classifier Based on Deep Convolutional Neural Network

In applications involving pattern recognition, convolutional neural networks, or CNNs, are frequently employed as supervised algorithms. Time-series data processing is a good fit for this neural network. Convolutional neural networks have been proposed as a method for classifying dangerous driving manoeuvres [42].

The input features of a CNN-based classifier to identify reckless or inattentive manoeuvre from aggressive driving behaviour are presented in Table 2, and the complete system is shown in Figure 5.

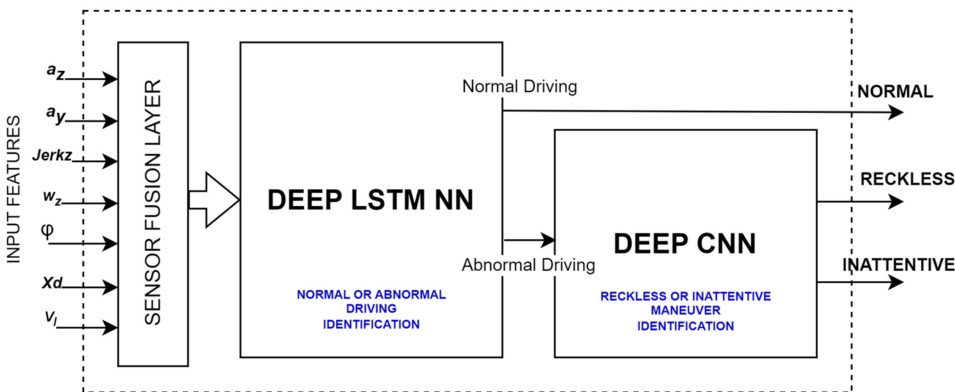

**Figure 5.** Proposed driving behaviour model using CNN.

The CNN is a deep learning neural network that consists of two convolutional layers that apply sliding convolutional filters to the input at hand and an input layer with seven neurons, one for each feature under consideration. There are 32 filters in the primary convolutional layer and 64 filters in the secondary layer. Furthermore, the inputs are left-padded in both convolutional layers to ensure that the outputs are the same length. In this experiment, a learning rate of 0.001 was employed. There are fifteen training epochs in all. Figure 6 depicts the architecture of the deep convolutional neural network module.

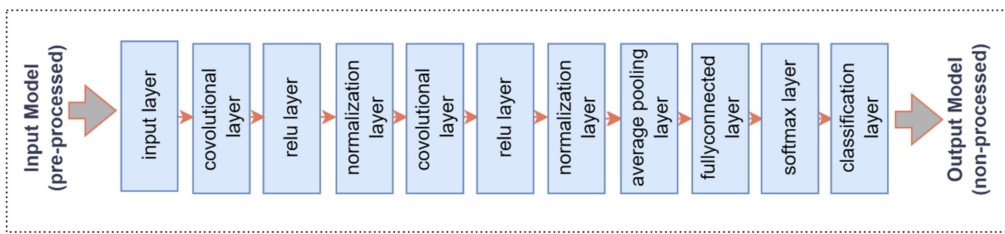

**Figure 6.** Convolutional neural network configuration.

*4.6. Classifier Based on Deep LSTM Neural Network*

Using the configuration depicted in Figure 2, the LSTM deep learning neural network implements the second module of the driving style identification system with an input layer with seven neurons. Table 2 lists the input features of the LSTM-based classifier, which is used to distinguish between aggressive driving behaviour and reckless or inattentive manoeuvres. Figure 7 illustrates the system.

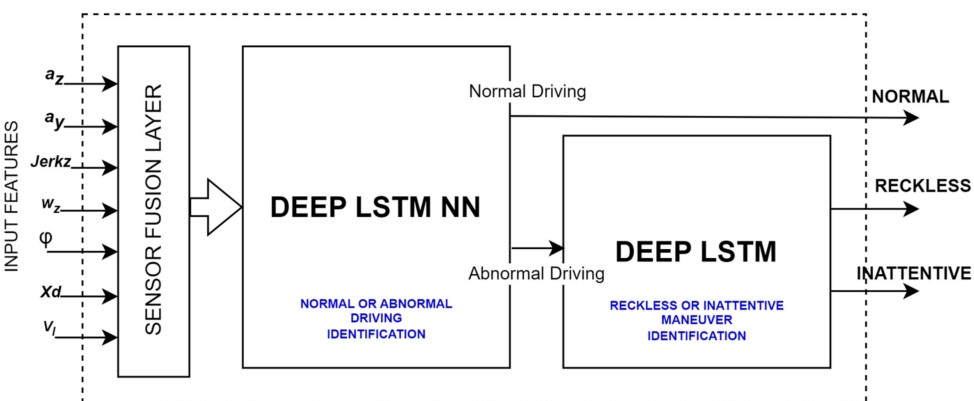

**Figure 7.** Proposed driving behaviour model using LSTM network classifier.

## 5. Results

### 5.1. First Stage: Classification between Normal and Anomalous Driving

This process is common for all the systems, whose second module will be implemented with different techniques. The analysis of the accuracy of the LSTM neural network identification model is based on the F1-score using Equations (16)–(19), which are used to assess the accuracy of the model [43].

$$F = \frac{2pr}{p + r} \tag{16}$$

where the expressions that define accuracy, *a*, precision, *p*, and recall, *r*, are the following.

$$a = \frac{TP + TN}{TP + FP + TN + FN} \tag{17}$$

$$p = \frac{TP}{TP + FP} \tag{18}$$

$$r = \frac{TP}{TP + FN} \tag{19}$$

TP stands for true positive, TN is true negative, FP is false positive, and FN is false negative. The confusion matrix is organized as shown in Table 4:

**Table 4.** Confusion matrix used in this work.

| Driver Profile | Predicted Positive | Predicted Negative |
|---|---|---|
| Current positive | TP | FP |
| Current negative | FN | TN |

Tables 5–7 show the obtained results. The first classifier utilizes the D1 driving profile as the training pattern, and the second classifier tested uses the 5-fold cross-validation method using all driving profiles. Tables 5 and 6 shows the confusion matrix which provides information on the actual (first column) and predicted (rows) performance classifications made by each identification model, with only D1 for training (D2 to D5 results) and with all the drivers' profiles (Table 6, 5 k-fold) for training. As the table shows, there is a good match between the true class and predicted class.

**Table 5.** Confusion matrix with the LSTM model for normal–abnormal driving estimation.

| D2 driver | Normal | Abnormal | D3 driver | Normal | Abnormal |
|---|---|---|---|---|---|
| Normal | 30,554 | 2462 | Normal | 31,275 | 2636 |
| Abnormal | 2231 | 10,583 | Abnormal | 2241 | 11,228 |
| **D4 driver** | Normal | Abnormal | **D5 driver** | Normal | Abnormal |
| Normal | 31,503 | 1654 | Normal | 28,029 | 1610 |
| Abnormal | 681 | 15,702 | Abnormal | 993 | 15,358 |
| **D6 driver** | Normal | Abnormal | | | |
| Normal | 27,284 | 1128 | | | |
| Abnormal | 1424 | 15,014 | | | |

The first proposed model, trained with D1, performs better than the 5 k-fold method, which uses 50–50% samples for training and testing. Results of these models are presented in Table 7. The best values have been bolded. As it is possible to see, they achieve high accuracy percentages and high F1-scores, indicating that the abnormal driving events are correctly identified.

**Table 6.** Confusion matrix with the LSTM model for normal–abnormal driving estimation.

| 5 k-Fold Strategy | Normal | Abnormal |
|---|---|---|
| Normal | 83,493 | 9068 |
| Abnormal | 17,415 | 27,634 |

**Table 7.** Result of the LSTM neural network classifier normal–abnormal driving.

| Driver | D2 | D3 | D4 | D5 | D6 | 5 k-Fold |
|---|---|---|---|---|---|---|
| Accuracy (%) | 89.76 | 89.71 | 95.29 | 94.34 | 94.31 | 81.88 |
| Precision (%) | 87.16 | 87.15 | 94.18 | 93.55 | 94.02 | 96.09 |
| Recall (%) | 87.57 | 87.79 | 95.43 | 94.24 | 93.68 | 78.06 |
| F1-score (%) | 87.36 | 87.46 | 94.75 | 93.88 | 93.85 | 78.79 |

In summary, the LSTM-based model is in fact capable of classifying normal and anomalous driving styles and therefore to identifies defective or risky manoeuvres vs. safe driving.

*5.2. Second Phase: Classification of Abnormal Types of Driving*

5.2.1. LSTM NN–HT Model

Table 8 shows the confusion matrix, which provides information on the actual and predicted values obtained by the Hotelling transform-based estimation system, when using D1 as the target and the rest of profiles for testing. The classifier is able to recognize the majority of the manoeuvres and therefore is able to detect and identify inattentive or reckless manoeuvres vs. aggressive driving among the abnormal driving. Based on the results in Table 9, the precision is high.

**Table 8.** Confusion matrix with the LSTM-HT classifier.

| D2 driver | Aggressive | Inattentive | D3 driver | Aggressive | Inattentive |
|---|---|---|---|---|---|
| Aggressive | 1810 | 84 | Aggressive | 1737 | 85 |
| Inattentive | 279 | 10,872 | Inattentive | 366 | 12,676 |
| **D4 driver** | Aggressive | Inattentive | **D5 driver** | Aggressive | Inattentive |
| Aggressive | 1759 | 62 | Aggressive | 1929 | 179 |
| Inattentive | 359 | 15,176 | Inattentive | 398 | 14,462 |
| **D6 driver** | Aggressive | Inattentive | | | |
| Aggressive | 2087 | 94 | | | |
| Inattentive | 431 | 13,530 | | | |

**Table 9.** Results of the LSTM-HT classifier aggressive–inattentive.

| Driver | D2 | D3 | D4 | D5 | D6 |
|---|---|---|---|---|---|
| Accuracy (%) | 97.21 | 96.96 | 94.46 | 96.59 | 96.74 |
| Precision (%) | 86.64 | 92.29 | 83.05 | 80.47 | 82.88 |
| Recall (%) | 95.56 | 95.33 | 96.59 | 91.50 | 95.69 |
| F1-score (%) | 95.10 | 98.38 | 92.46 | 93.58 | 92.82 |

Driving profiles D2 and D3 achieve better classification performance in terms of accuracy and F1 score. This may be due to the fact that the driving profiles were obtained using a diesel vehicle, the same as D1 that was the driving profile used for training. The technical characteristics of petrol, diesel and electric vehicles are not exactly the same.

### 5.2.2. LSTM NN-CNN Model

For the LSTM-CNN system, Table 10 shows the confusion matrices, which provide information on the actual and predicted values obtained with the CNN-based estimation system, when using D1 as the target and the rest of profiles for testing. Table 11 presents the results obtained in the classification of abnormal manoeuvres. It can be seen by comparing them that defective and inattentive conduction events are recognized but most data samples have a medium accuracy percentage and a poor F1-score.

**Table 10.** Confusion matrix with the LSTM-CNN classifier.

| **D2 driver** | Aggressive | Inattentive | **D3 driver** | Aggressive | Inattentive |
|---|---|---|---|---|---|
| Aggressive | 91 | 4934 | Aggressive | 115 | 4959 |
| Inattentive | 281 | 7739 | Inattentive | 337 | 8453 |
| **D4 driver** | Aggressive | Inattentive | **D5 driver** | Aggressive | Inattentive |
| Aggressive | 94 | 6418 | Aggressive | 174 | 5596 |
| Inattentive | 336 | 10,508 | Inattentive | 399 | 10,799 |
| **D6 driver** | Aggressive | Inattentive | | | |
| Aggressive | 143 | 5578 | | | |
| Inattentive | 380 | 10,041 | | | |

**Table 11.** Results of the LSTM-CNN classifier aggressive–inattentive.

| **Driver** | **D2** | **D3** | **D4** | **D5** | **D6** |
|---|---|---|---|---|---|
| **Accuracy (%)** | 60.02 | 61.80 | 61.08 | 64.67 | 63.09 |
| **Precision (%)** | 42.77 | 42.23 | 41.97 | 48.12 | 45.86 |
| **Recall (%)** | 49.15 | 49.21 | 49.17 | 49.72 | 49.42 |
| **F1-score (%)** | 39.08 | 40.15 | 39.19 | 42.88 | 40.85 |

### 5.2.3. LSTM-LSTM Model

Applying the LSTM-LSTM driving style identification system, Table 12 represents the confusion matrix, which provide information on the actual and predicted values obtained with the LSTM-based classifier, when using D1 as the target and the rest of drivers' profiles for testing. Table 13 shows the results obtained in the classification of abnormal manoeuvres. It can be seen that the aggressive and inattentive patterns are recognized, but results achieve a medium accuracy percentage and poor F1-score.

**Table 12.** Confusion matrix with the LSTM-LSTM classifier.

| **D2 driver** | Aggressive | Inattentive | **D3 driver** | Aggressive | Inattentive |
|---|---|---|---|---|---|
| Aggressive | 92 | 4933 | Aggressive | 115 | 4959 |
| Inattentive | 282 | 7738 | Inattentive | 339 | 8451 |
| **D4 driver** | Aggressive | Inattentive | **D5 driver** | Aggressive | Inattentive |
| Aggressive | 95 | 6417 | Aggressive | 174 | 5596 |
| Inattentive | 337 | 10,507 | Inattentive | 392 | 10,806 |
| **D6 driver** | Aggressive | Inattentive | | | |
| Aggressive | 142 | 5579 | | | |
| Inattentive | 377 | 10,044 | | | |

**Table 13.** Results of the LSTM-LSTM classifier aggressive–inattentive.

| Driver | D2 | D3 | D4 | D5 | D6 |
|---|---|---|---|---|---|
| **Accuracy (%)** | 60.02 | 61.78 | 61.08 | 64.71 | 63.10 |
| **Precision (%)** | 42.83 | 44.18 | 42.03 | 48.31 | 45.82 |
| **Recall (%)** | 49.16 | 49.20 | 49.18 | 49.75 | 49.43 |
| **F1-score (%)** | 39.10 | 40.15 | 39.20 | 41.89 | 40.84 |

## 6. Discussion of the Results

The study of driving behaviour is necessary but complex. The impact of different internal and external factors on driver's behaviour is difficult to analyse. While much research has been dedicated to understanding the impact of some driving requirements or specific traffic environments on driving performance, there is limited information on how drivers adapt their driving behaviour and on what they based their driving style.

The results showed previously prove that there are many variables that can be used to estimate the driver's behaviour. Machine learning techniques can be used to classify a specific style of driving. Among the different designs proposed in this work, the LSTM NN–HT estimation model is in fact able to classify most driving styles on the road and therefore to identify risky manoeuvres vs. what can be considered normal and thus safe driving. The proposed system achieves a high degree of accuracy classifying aggressive and inattentive manoeuvres, as shown by the values of the accuracy and F1-score metrics. In contrast, other models tested based on deep neural networks perform significantly worse.

Indeed, the methodology based on an LSTM neural network and the Hotelling transform achieves an F1-score of up to 94.75%. This model has been designed through two stages, the first one to identify normal or safe manoeuvres, and the second one distinguishes between aggressive or inattentive manoeuvres. In comparison with other studies found in the literature, in [16], the authors propose a solution to the driver behaviour classification problem based on an LSTM-FCN network that detects whether a driving session involves aggressive behaviour. The problem is formulated as a time series classification, and the validity of the approach is tested on the same database as this work, UAH-DriveSet. The proposed system achieves an F1-score of up to 95.88%, but only aggressive driving behaviour is taken into account, and it does not include road geometry. In [44], the model for driver behaviour classification is based on stacked LSTM recurrent neural networks. The same UAH-DriveSet database is used in which the three driving classes of this study are distinguished. The proposed Stacked-LSTM obtains an F1 measure performance of 91%. It also does not include the influence of the geometric characteristics of the road, and the one proposed here obtains a 98.38% F1-score.

The system proposed in this work, which combines those two techniques, can help users to drive more safely and comfortably on the roads by means of real-time measured features of smartphones and GPS. In this way, better and more appropriate driving can be achieved on these roads, thus reducing the risk of accidents.

## 7. Conclusions

This research proposes a neural network-based driving behaviour recognition model that is employed to recognize driving manoeuvres that may suggest a risky driving behaviour on both regular two-lane roads and highways.

The model consists of two modules, one based on deep long-short term memory neural network to identify abnormal manoeuvres, and the other uses deep neural networks and the Hotelling transform method to distinguee between aggressive and inattentive driving. It takes as inputs actual accelerometer, gyroscope and GPS readings from a vehicle trip. With the help of these measures, which are easily available even in affordable smartphones, it is feasible to determine the acceleration caused by the geometry of the road and the acceleration perceived by the user, taking into consideration the impact of some geometric characteristics of the road on the driving. This information can be used to detect defective, distractive or inattentive driving manoeuvres while driving, considering the effect of the

geometry of the road. Therefore, the main contribution of this work is that it allows the incorporation of indirectly estimated information about the state of the road and its geometry to identify risky manoeuvres. This information affects the way that you drive, allowing for better identification of different driving profiles.

The findings from a publicly available dataset of drivers with six distinct driving profiles—each with unique characteristics—allow for the drawing of certain conclusions. First, all driving profiles with uncommon movements have been accurately detected using this model. Different driving styles (aggressive, sleepy and normal) were also considered. In fact, normal driving is defined differently depending on the driver. Second, all driving profiles with inattentive or reckless driving have also been rightly identified from samples containing abnormal driving. The LSTM-HT-based system has the best performance in this task because it achieves high accuracy. Indeed, it achieves an accuracy of 96.46% and 97.21% and a F1 score performance between 92.46% and 98.38% when trained with the D1 driving profile.

The results of this driving identification model are intriguing and practical; they could be a supplement to other in-car driver support systems that are applicable to self-driving cars.

Several promising future works can be addressed. Different and additional characteristics of the road could be explicitly included, such as road curvature or the speed of other cars traveling on the same road section. Furthermore, several additional variables, like wheel interaction with the road surface and group permeability to faster cars, are beginning to be recorded by smart sensors and may be used as inputs for further models. In addition, it would be also interesting to complement the system including external environmental factors that may influence driving behaviour such as weather or even with vehicles or driver's characteristics, such as in [1].

**Author Contributions:** F.B. proposed the conceptualization, methodology, data calculation and result analysis. M.S. and M.R. contributed to the conceptualization, the review and discussion. All authors have read and agreed to the published version of the manuscript.

**Funding:** This research received no external funding.

**Data Availability Statement:** All the data used in this research are obtained from the formulas or from the cited public database.

**Conflicts of Interest:** The authors declare no conflicts of interest.

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
