# Peer review of "Driving Behaviour Estimation System Considering the Effect of Road Geometry by Means of Deep NN and Hotelling Transform"

_electronics, doi:10.3390/electronics13030637_

Round 1

Reviewer 1 Report

Comments and Suggestions for Authors

1. Equation (4) appeared twice, so that subsequent equations need to be renumbered.

2. Equation (5) has an error, Jerk should be the derivative of acceleration over time.

3. Please check if the expressions of equations (6) and (7) are correct.

4. Where do equations (6) and (7) come from, and is there any literature supporting these equations? In my opinion, in the absence of evidence, these equations seem to be less rigorous.

5. The data in Table 1 seems to be continuously recorded, so how are the samples divided? How many training and testing samples were used by the author in constructing their network model?

Author Response

Reviewer 1

We want to thank the reviewer for the insightful and useful comments that have help improved our work. Changes in the paper are highlighted.

Comments and Suggestions for Authors

  1. 1. Equation (4) appeared twice, so that subsequent equations need to be renumbered.

Thank you for detecting it. The numbering of the equations has been reviewed and corrected.

  1. 2. Equation (5) has an error, Jerk should be the derivative of acceleration over time.

You are right. Corrected. Thank you.

  1. 3. Please check if the expressions of equations (6) and (7) are correct.
  2. Where do equations (6) and (7) come from, and is there any literature supporting these equations? In my opinion, in the absence of evidence, these equations seem to be less rigorous.

Thank you for bringing this out. The expressions, now numbered (7) and (8), are correct. They are based on the fact that a different assessment of driving established as normal is outside expected values of the variables that measure the stability of the longitudinal and lateral control of the vehicle, in this case longitudinal and lateral accelerations [45]. Metrics that are mainly used to specifically assess comfort and safety during driving manoeuvres are the mean and standard deviation of variables such as acceleration or jerk because humans are more sensitive to rapid changes in acceleration, as stated in [45].

This brief explanation has been added in the revised version of the manuscript.

  1. The data in Table 1 seems to be continuously recorded, so how are the samples divided? How many training and testing samples were used by the author in constructing their network model?

Thank you for this comment. We think we have expressed ourselves poorly, and we have improved the explanation of this point in the text. On the one hand, we have added a further description of the data of the database and how they were recorded. Besides, we have also added that there are a total of 275,220 samples that contain all the drivers’ profile, of which 41,630 belong to driver D1. The data were initially randomly divided into 50% for training and the other 50% for testing, making use of all driving profiles. To improve the models, other experiments were carried out in which the driving profile D1 was used to train the model and the rest of the profiles for testing. The results showed that the best classification was obtained using the training data of profile D1.

This information has been added in the revised version of the manuscript.

Reviewer 2 Report

Comments and Suggestions for Authors

This paper proposed a Driving behavior estimation system. This paper is relatively well written and clear. The flow and representation are good. Nevertheless, some improvements could be made before the publication. A few issues need to be addressed.

1. The novelty of this work needs to be further clarified.

2.In the abstract, it should be further elaborated on how to integrate road geometry into driving behavior estimation.

3. Is the method for determining driving behavior accurate?What is the basis for defining the scope of different driving behaviors?

4.What are the advantages of using a combination of LSTM and PCA methods for driving behavior estimation compared with othersThe basis for selecting LSTM and PCA separately should be further clarified.

5.The third part of the formula label is wrong.

6.The classification standard of driving behavior is not clearly defined, and the classification standard of how to judge whether a driving behavior is abnormal or whether it belongs to reckless driving or distracted driving needs to be further clarified.

7." The influence of the road geometry can be taken into account indirectly through the vehicle dynamics" proposed in the paper lack of theoretical support.

8.The source of training and test data has not been described in detail, nor has the actual effect of the model been verified in the intelligent driving system, so it is difficult to evaluate the generalization ability of the model.

9.The paper only considers the vehicle dynamics characteristics, without considering the influence of external environmental factors on driving behavior, which affects the accuracy of the model to a certain extent.

10.The result part gives a complete key index of the proposed classifier, whether the safety and comfort index can be given as the abstract mentioned.

11.In section 4.1, the first module has four input features, which are inconsistent with the model given in the subsequent pictures.

12.What is the relationship between Hotelling Transform and PCA, HT is mentioned in the title, but PCA is explained at great length in the second part and the Classifier based on Hotelling Transform (HT) section.

13.Functionally, both levels of the model have classification capabilities and both recognize driving behaviors, so why not use a network? Whether the structure in the paper will cause the time out-of-sync problem of behavior classification.

14.What is the time range for neural network input data? What impact is brought about by different ranges of prediction accuracy?

Comments on the Quality of English Language

Minor editing of English language required

Author Response

Reviewer 2

  1. The novelty of this work needs to be further clarified.

Thank you for this comment that helps to value our work. The novelty of this work has been highlighted in the introduction and conclusions.

In our opinion, the main contribution consists of incorporating the geometry of the road as another variable to evaluate its influence on the way of driving, or on a manoeuvre, something that is not usually taken into account. Furthermore, it is done indirectly but through real inertial measurements taken by low-cost devices, with which the acceleration perceived by the vehicle's passengers is obtained without the need to have direct information on the actual state of the infrastructure.

2.In the abstract, it should be further elaborated on how to integrate road geometry into driving behaviour estimation.

Thank you. The abstract has been further elaborated to explain how the road geometry is integrated into de driving behaviour.

  1. Is the method for determining driving behaviour accurate?What is the basis for defining the scope of different driving behaviours?

Thank you for the question. The basis for defining the scope of different driving behaviours is the information extracted through inertial sensors of a smartphone. These signals come from a database used in the scientific literature, UAH-DriveSet [31], which characterizes various driving behaviours according to acceleration, speed, steering, …. The signals have been filtered and processed to be as accurate as possible. That said, driving behaviour will always be subjective and with a certain degree of uncertainty.

4.What are the advantages of using a combination of LSTM and PCA methods for driving behaviour estimation compared with others. The basis for selecting LSTM and PCA separately should be further clarified.

Thanks for the question. A system has been designed that classifies in cascade, first between normal and abnormal behaviour, and then classifies abnormal forms of driving as aggressive or inattentive. LSTM networks have proven to be very effective for working with time series, which is why they have been applied for the first phase with good results. For the second, which is a supervised classification problem, various techniques have been applied, and the hybrid configuration that has given the best results has been that of LSTM- HT.

5.The third part of the formula label is wrong.

Thank you for spotting this. Formulas have been revised and corrected.

6.The classification standard of driving behaviour is not clearly defined, and the classification standard of how to judge whether a driving behaviour is abnormal or whether it belongs to reckless driving or distracted driving needs to be further clarified.

Thank you for this comment. In the manuscript, some paragraphs have been added to explain what is considered abnormal driving behaviour, according to some measures proposed by [45], and normal, distracted, reckless and aggressive driving style, as defined by [17].

7." The influence of the road geometry can be taken into account indirectly through the vehicle dynamics" proposed in the paper lack of theoretical support.

Thank you for this comment. At the end of section 3 it has been clarified with the support of the mathematical formulas given in references [27-30].

8.The source of training and test data has not been described in detail, nor has the actual effect of the model been verified in the intelligent driving system, so it is difficult to evaluate the generalization ability of the model.

We are sorry for not having described the materials used well enough. We have added a more detailed description of the information of the UAH-DriveSet database [31]. This is a driving database with inertial measurements and images while traveling on a conventional road located in Madrid (Spain) with different vehicles. We have also added information about the training and testing data used from that set.

9.The paper only considers the vehicle dynamics characteristics, without considering the influence of external environmental factors on driving behaviour, which affects the accuracy of the model to a certain extent.

You are right, in fact there are other works that do consider the influence of external environmental factors on driving behaviour, such as the weather or the state of the car [45]. In this case they are data-driven models, based only on the measured data of the vehicle dynamics and there is not any other complementary information. We think it is a suggestive proposal for future work and it has been included in the final section of the article.

10.The result part gives a complete key index of the proposed classifier, whether the safety and comfort index can be given as the abstract mentioned.

Thanks for this comment. The abstract now mentions some of the results obtained with the system and specifies that the detection of a dangerous manoeuvre allows the driver to generate aid that would make the trip safer.

11.In section 4.1, the first module has four input features, which are inconsistent with the model given in the subsequent pictures.

You were right. Accordingly, figure 4 has been changed to keep the coherence regarding the number of inputs. Thank you.

12.What is the relationship between Hotelling Transform and PCA, HT is mentioned in the title, but PCA is explained at great length in the second part and the Classifier based on Hotelling Transform (HT) section.

Thank you for the question. The PCA approach converts the original space input information to the PCA processed space, only with its uncorrelated information [36]. This processing is done by the Hotelling matrix. In the manuscript, to avoid any confusion, we have now explained that we apply the HT and the methodology followed to apply this transform has been described in detailed.

  1. Functionally, both levels of the model have classification capabilities and both recognize driving behaviours, so why not use a network? Whether the structure in the paper will cause the time out-of-sync problem of behaviour classification.

Thank you for this discussion, which we find very interesting. Indeed, a single multiclass neural network could be applied. But the number of errors would be greater since there are not many examples of aggressive or distracted behaviour compared to normal, so the sets would be unbalanced. The cascade configuration of one vs rest classification is very common and useful in different problems. That said, the second module can be a network, as it has been tested in the paper using both, a second LSTM and a CNN, but in both cases the hybrid system has given worse results than with the LSTM-HT configuration. Finally, it is possible in a one vs rest classification that there may be a certain delay in the classification of the behaviour, as the reviewer rightly comments. We understand that in this case a real-time implementation of the system should be assured.

14.What is the time range for neural network input data? What impact is brought about by different ranges of prediction accuracy?

Thank you for your question. We find this comment very interesting if we want to recognize anomalous behaviour in real time. Once the network has been trained for a particular driver, it depends on the manoeuvre whether the system is able to identify abnormal behaviour. For example, the manoeuvres associated with distractions are usually followed with a rapid driver reaction to correct vehicle position, as a distracted driving style has an instantaneous and irregular nature, while the characteristics of aggressive or drunk driving are maintained over time and present a periodic pattern of misbehaviour.

Having said that, in our system we use all the data associated to each driver as inputs of the neural network.

Reviewer 3 Report

Comments and Suggestions for Authors

This study aims to propose an intelligent prediction model to identify dangerous or careless driving maneuvers on the roads.

Sections 1 and 2 comprehensively introduce the topic, objective and background state of the art of the proposed article. However, the reviewer suggests increasing the bibliometric references, not only in Section 1 Introduction but also in Section 2. The proposed study aims to evaluate the effect of road geometry on driver driving behavior through intelligent systems that analyze vehicle dynamic variables. For example, the reviewer suggests that the literature references could be enriched with some studies, that through the study of operating speeds, have analyzed the influence of road geometry and the entire infrastructure on user behavior and road safety. The reviewer suggests consulting the following recent studies:

https://doi.org/10.1016/j.trpro.2023.02.227

https://doi.org/10.3390/infrastructures8020030

The reviewer suggests increasing the quality of resolution of all figures, especially to aid reading of the lettering.

The reviewer suggests using Section 6 Discussion to summarize and comment on the proposed model and the results obtained. In addition, the results could be compared with those of other studies.

The reviewer suggests that Section 7 Conclusions also include the results in numerical form.

Comments on the Quality of English Language

The quality of writing in this manuscript is sufficient, so moderate editing of the adopted English is recommended.

Author Response

Reviewer 3

  1. 1. Sections 1 and 2 comprehensively introduce the topic, objective and background state of the art of the proposed article. However, the reviewer suggests increasing the bibliometric references, not only in Section 1 Introduction but also in Section 2. The proposed study aims to evaluate the effect of road geometry on driver driving behavior through intelligent systems that analyze vehicle dynamic variables. For example, the reviewer suggests that the literature references could be enriched with some studies, that through the study of operating speeds, have analyzed the influence of road geometry and the entire infrastructure on user behavior and road safety. The reviewer suggests consulting the following recent studies:

https://doi.org/10.3390/infrastructures8020030

https://doi.org/10.1016/j.trpro.2023.02.227

Thank you for the suggested references. The bibliographic review has been improved by making use of these sources, which we consider recent and interesting.

2.The reviewer suggests increasing the quality of resolution of all figures, especially to aid reading of the lettering.

Thank you for this comment. The resolution of some figures has been improved.

3.The reviewer suggests using Section 6 Discussion to summarize and comment on the proposed model and the results obtained. In addition, the results could be compared with those of other studies.

Thanks for this suggestion that has enriched our work. In section 6 a summary of the proposed methodology and the results obtained has been added. Furthermore, the results have been compared with some works found in the literature that use the same database of drivers.

4.The reviewer suggests that Section 7 Conclusions also include the results in numerical form.

Thank you very much for this comment that improves that section. The results have been included in numerical form in the conclusions.

Round 2

Reviewer 1 Report

Comments and Suggestions for Authors

All my concerns have been resolved.

Reviewer 2 Report

Comments and Suggestions for Authors

The article has been well revised and can be accepted in present form for publication.

Comments on the Quality of English Language

Minor editing of English language required